# An Italian Twin Study of Non-Cancer Chronic Pain as a Wide Phenotype and Its Intensity

**DOI:** 10.3390/medicina58111522

**Published:** 2022-10-26

**Authors:** Corrado Fagnani, Virgilia Toccaceli, Michael Tenti, Emanuela Medda, Maurizio Ferri, Maria Antonietta Stazi, William Raffaeli

**Affiliations:** 1Centre for Behavioural Sciences and Mental Health, Istituto Superiore di Sanità (Italian National Institute of Health), 00161 Rome, Italy; 2Fondazione ISAL—Institute for Research on Pain, 47921 Rimini, Italy

**Keywords:** chronic pain, pain intensity, wide phenotype, twin study, heritability, genetics, environmental influences

## Abstract

*Background and Objectives*: Non-cancer chronic pain (CP) results from the interaction between genetic and environmental factors. Twin studies help to estimate genetic and environmental contributions to complex traits such as CP. To date, twin studies on the heritability of pain phenotypes have relied almost exclusively on specific diagnoses, neglecting pain intensity. This study aims to estimate the genetic and environmental contributions to CP occurrence as a wide phenotype and its intensity among a non-clinical population. *Materials and Methods*: A nationwide online survey was conducted in February 2020 on 6000 adult twins enrolled in the Italian Twin Registry. A five-item questionnaire, designed and validated by our study group, was administered to detect the CP condition along with its intensity, underlying causes or triggers, treatments, and self-perceived efficacy. The twin study design was used to infer the relative weight of genes and environment on CP occurrence and intensity, and biometrical modelling was applied to these phenotypes. *Results*: A total of 3258 twins, aged ≥18, replied to the online survey (response rate 54%). These included 762 intact pairs (mean age: 39 years; age range: 18–82 years; 34% male; CP prevalence: 24%), of whom 750 pairs were subjected to biometrical modelling after the exclusion of pairs with either unknown zygosity or cancer-associated CP. Broad-sense heritability estimates were driven by non-additive genetic effects and were 0.36 (0.19–0.51) for CP occurrence and 0.31 (0.16–0.44) for CP intensity. No evidence emerged for either sex differences in genetic and environmental variance components or interactions of these components with age. *Conclusions*: Moderate non-additive genetic components were suggested for non-cancer CP occurrence and its intensity. These results encourage further research on the gene–gene interactions underlying CP liability and associated phenotypes, and also strengthen the need for prevention strategies to avoid CP occurrence or to decrease pain intensity.

## 1. Introduction

Non-cancer CP (referred to as CP in this work), i.e., pain persisting or recurring for more than three months [1], affects nearly 20% of individuals in western countries [2,3], with detrimental effects on their functioning and quality of life [4]. CP, in fact, leads to several biological, physical, and psychosocial consequences contributing to the load for patients, health care systems, and societies [5]. Increasing evidence points to CP as a disease in its own right [6] which represents a serious challenge to health care providers due to its complex multifactorial pathophysiology [7]. According to the Diathesis–Stress Model, CP results from a complex interaction between genetic predisposition (“Diathesis”) and multiple environmental factors (“Stress”) [7]. Twin studies, in particular, are extremely useful for estimating genetic and environmental contributions to complex traits such as CP. By comparing trait concordance in genetically identical monozygotic (MZ) twins with that in dizygotic (DZ) twins (who share on average 50% of genes), and assuming that relevant environmental exposures are shared by MZ twins to the same extent as DZ twins (“equal environments assumption”), it is in fact possible to estimate the relative weight of genetic and environmental factors in the occurrence of the trait [8]. As reviewed by Nielsen and colleagues [9], available data from twin studies on different pain phenotypes indicate an average heritability of 50% for migraine, tension-type headaches and chronic widespread pain, 35% for back and neck pain, and 25% for irritable bowel syndrome. However, these results were affected by a high variability in phenotype definitions. Moreover, twin studies on the heritability of pain phenotypes have almost exclusively relied on dichotomous diagnoses, thus neglecting pain intensity as a continuous phenotype. This aspect was highlighted as a major weakness of previous studies, not only in relation to a loss of statistical power, but also with respect to the potential confounding effect of pain intensity on the diagnosis of the pain syndrome, given that patients with more severe pain might be more likely to be diagnosed than those with milder pain. In this context, it is difficult to establish whether and to what extent the estimated genetic effects reflect pain processing or the tissue pathology causing the pain [9].

Despite the fact that CP may actually refer to different phenotypes and diagnostic entities recently categorized within the International Classification of Diseases-11 (ICD-11) [10], there are several reasons why, in this research field, CP should be considered as a wide phenotype, i.e., as a general affliction occurring in many different conditions and across diagnoses. Some CP syndromes, in fact, seem to show an overlap in the presenting symptomatology and to share a common genetic predisposition [11], while the major comorbidities, outcomes, and psychosocial burden of CP are frequently more dependent on the severity and interference of pain than on the diagnosis in itself [12].

In this framework, the main aim of the present study is to estimate the genetic and environmental components of CP occurrence as a wide phenotype (irrespective of underlying causes/triggers) and its intensity, as well as to explore the possible sex- and age-moderation of these components in a sample of adult twins enrolled in the Italian Twin Registry (ITR) of the Istituto Superiore di Sanità.

The present study was approved by the Ethics Board of the Istituto Superiore di Sanità on 14 April 2020. Participants underwent an informed consent procedure.

## 2. Materials and Methods

### 2.1. Recruitment of Study Sample

A nationwide online survey, using the LimeSurvey Professional Platform (LimeSurvey GmbH, Hamburg, Germany), was conducted in February 2020 on 6000 MZ and DZ adult twins of Caucasian origin previously enrolled in the population-based ITR [13]. Currently, the ITR contains socio-demographic and health information on more than 30,000 twins of all ages and geographical areas, and it promotes genetic–epidemiological research, particularly on behavioral and mental health traits [14]. Detailed information on survey procedures and tools can be found elsewhere [15].

### 2.2. Measures

#### 2.2.1. Zygosity Assessment

Twin pairs were classified as MZ or DZ based on standard questions on physical resemblance and frequency of confusion of the twins by family members and strangers [16]. This is a well-known method in twin studies, which has been reported to be generally over 95% accurate.

#### 2.2.2. CP Assessment

The Brief Five-Item Chronic Pain Questionnaire was used for CP assessment. It is a validated brief self-administered measure, particularly suitable to detect persistent states of pain and related characteristics in general population surveys [17]. It is composed of a first filtering item able to discriminate between CP “affected” and “non-affected” individuals. Among CP sufferers, the remaining four items concern CP intensity, possible underlying CP causes/triggers, any drugs/treatments and their frequency, and the self-perceived effectiveness of treatments. In this study, a body mannequin to map CP areas was also administered. The Brief Five-Item Chronic Pain Questionnaire showed good content and construct validity, as well as adequate test–retest reliability in the Italian general population [17]. The questionnaire is not a psychometric scale, and therefore it has no total score or cut-offs. For the purposes of the present study, the two items regarding CP occurrence (dichotomous scale: 0 = no, 1 = yes), based on the International Association for the Study of Pain (IASP) definition of CP [1], and CP intensity (ordinal scale: 1 = very mild, 2 = mild, 3 = moderate, 4 = severe, 5 = very severe) were subjected to biometrical structural equation modelling (see below), while the remaining questionnaire items were considered only in the descriptive analyses. As regards body maps, clinical experts are currently evaluating pain sites to detect relevant macro areas and patterns that may help resolve the highly fragmented information; these data will be presented in later reports.

### 2.3. Statistical Analyses

#### 2.3.1. Sample Description

The socio-demographic and CP-related characteristics of the study sample were summarized using means (with standard deviations and ranges) for continuous variables and percentages for categorical variables. Descriptive analyses were performed by the Stata software version 16 (StataCorp LLC, College Station, TX, USA).

#### 2.3.2. Biometrical Structural Equation Modelling

Biometrical structural equation model-fitting analyses were conducted, in accordance with the assumptions of the twin design, to estimate the heritability and the environmental component of CP occurrence and intensity.

For the analysis of CP intensity, the following scale was considered: 0 = no pain, 1 = very mild/mild, 2 = moderate, 3 = severe/very severe, with the “very mild” and “very severe” responses of the original scale combined, respectively, with the “mild” and “severe” ones due to low numbers. As in other commonly used scales of pain intensity (e.g., the visual analogue scale (VAS), the numerical rating scale (NRS)), two endpoints from “no pain” to “very severe pain” were considered.

Given the dichotomous and ordinal nature of occurrence and intensity respectively, liability-threshold models were applied [8]. Using this approach, within-pair (tetrachoric or polychoric) correlations for the targeted phenotypes were estimated in the MZ and DZ pairs to preliminarily assess the relative contributions of genes and environment to trait liabilities based on MZ vs. DZ correlations. Furthermore, models including additive genetic (A), either non-additive genetic (D) or shared (familial) environmental (C), and unique (individual-specific) environmental (E) components, with sex and age as covariates, were fitted. The A component represents the additive effects of all gene variants (i.e., alleles) that influence the trait, without interactive effects; the D component represents interactions between alleles at the same fixed chromosomal site (i.e., locus) (“dominance”) or at different loci (“epistasis”); the C component represents the effects of environmental factors that were shared by the twins within the family, particularly during childhood and adolescence (e.g., rearing environment, family socio-economic status, parental behaviors, etc.), or those that were shared in the womb (e.g., hormonal exposures); the E component represents the effects of environmental factors that are unique to an individual (e.g., lifestyles, relations with peers, infections, etc.), including measurement error [8]. The ADE and ACE models were compared with one another based on the Akaike information criterion (AIC), and parameter estimates were derived under the best-fitting model (i.e., the one with the lowest AIC). To test for possible sex-related heterogeneity in the etiological sources, correlation and model-fitting analyses were replicated on five zygosity-by-sex groups (i.e., MZ male, MZ female, DZ male, DZ female, DZ unlike sex), with only age as a covariate. The possible interaction of genetic and environmental components with age was also tested by including age as a continuous moderator in the models for the MZ and DZ groups, based on the parameterization of Purcell’s G×E model for categorical data [18]. Model-fitting was carried out with the Mx software [19].

## 3. Results

### 3.1. Participants

Among the 6000 invited twins, 3258 replied to the online survey (response rate 54%). This response rate was substantially higher than that generally observed in previous ITR surveys. No significant differences emerged between the respondents and non-respondents with respect to the main socio-demographic characteristics (i.e., age, sex, education). Among the 3258 twins, there were 1524 twins from 762 intact pairs, plus 1734 unmatched twins. Given that correlation and model-fitting analyses of twin data mainly exploit within-pair information, only twins from the 762 intact pairs were considered. No major differences were detected between twins from intact pairs and unmatched twins with respect to zygosity or to socio-demographic and CP-related profiles. All 762 pairs were included in the descriptive analysis, while 750 pairs included in the correlation analysis and structural equation modelling, after the exclusion of 7 pairs with unknown zygosity and 5 pairs for which CP occurrence was declared to be associated with a tumor diagnosis.

### 3.2. Sample Description

The main socio-demographic and CP-related characteristics are summarized in Table 1.

The mean age was 39 years (range 18–82 years), 34% of the twins were male, and 57% were MZ. Their education level was medium–high, with more than 50% of subjects having at least a three-year university degree. CP prevalence was about 24%, with 60% of CP-affected subjects reporting a moderate pain intensity and 18% declaring a severe or very severe pain. As regards the causes of pain, trauma without surgery was the most reported origin (40%), followed by “another illness diagnosed by a doctor” (35%) and a “poorly defined disease” (21%). Among twins suffering from CP, 74% reported being under (regular, cyclic, or sporadic) treatment, and of these, 86% declared that they had received beneficial medication effects.

### 3.3. Biometrical Structural Equation Modelling

The results of the correlation and model-fitting analyses are shown in Table 2.

For both CP occurrence and intensity, within-pair MZ correlations were remarkably higher than DZ correlations, with the latter being of vanishingly low magnitude. A similar pattern was observed when stratifying the sample by zygosity and sex; however, a strong random variability prevented the easy interpretation of the estimates, in particular for a higher (yet not significantly higher) correlation in unlike-sex DZ twins compared with same-sex DZ twins. The correlation pattern did not suggest a role for shared (familial) effects on trait liabilities, but did suggest a key role for non-additive (possibly epistatic) genetic influences. Accordingly, the ADE structural equation models (i.e., models including A, D, and E components—see Methods, above) provided a better description of the data than the ACE models (i.e., models including A, C, and E components—see Methods, above). Under these models, for both CP occurrence and intensity, a negligible additive genetic contribution was found in the sex-unstratified analysis, and in both males and females. Broad (i.e., A + D) heritability estimates were basically driven by non-additive genetic effects, with values in the two-group analysis of 0.36 (A = 0) and 0.31 (A = 0.02) for CP occurrence and intensity, respectively. No sex differences were detected in the genetic and environmental variance components (*p* = 0.91 for CP occurrence; *p* = 0.87 for CP intensity). Furthermore, no evidence emerged for interactions of these components with age.

## 4. Discussion

Although aware of the complexity of the phenomenon [20], we have contributed to the knowledge of the etiology of CP as a wide phenotype, beyond the specific definitions adopted in previous twin studies, which were mostly conducted in clinical or experimental settings [9]. In fact, we considered a large non-clinical sample of twins enrolled in the population-based ITR, surveyed at the beginning of 2020, just before the spread of the COVID-19 pandemic. We followed the line of investigation of other authors who had exploited the resource of twin registries from around the world. Among these, Burri et al. [21] analyzed the wide phenotype of CP only in females, while our Italian sample also included a male component, thus providing more generalizable findings. We are aware that different CP conditions may have different genetic influences [22]. However, it has been suggested that several CP phenotypes may have a common genetic background [9,23,24]. For instance, Vehof et al. [11], using a large cohort of twins (N = 8564), suggested that different CP conditions, such as chronic widespread musculoskeletal pain, chronic pelvic pain, migraine, dry eye disease, and irritable bowel syndrome, may find a common ground based on their phenotypic relationship with an underlying latent factor, whose heritability was estimated to be as high as 0.66. A recent large-scale human genetic study has also shown that many different pain phenotypes have positive and significant genetic correlations with each other. This indicates that common genetic risk factors confer liability to pain at many different sites across the body, suggesting shared risk factors and, potentially, disease mechanisms [25]. Therefore, gathering CP phenotypes could help to produce further knowledge about shared genetic mechanisms underlying different CP syndromes, particularly for genetic association studies where sample size is a key factor for success.

We have contributed to the knowledge regarding the twin-based heritability of CP by considering not only the dichotomous occurrence of pain but also, and most importantly, its intensity level defined as an ordinal variable. This may represent a valuable step forward in the process of overcoming the limitations imposed by the simple definition of CP as a binary trait and moving towards the analysis of CP assessed on a more refined, possibly continuous scales of intensity [9]. In this respect, it is noteworthy that CP occurrence and intensity (as defined in our study) appeared to have similar genetic and environmental components, with around one third of individual differences in both traits explained by genetic factors, and the remaining two thirds being due to individual-specific environmental factors. In other words, according to our data, the etiological architecture of CP onset seemed invariant when the information on the simple disease occurrence was refined with the intensity level (i.e., in terms of liability-threshold modelling, when the one-threshold occurrence-based model was turned into a three-threshold intensity-based model, with the base category of “no pain” remaining unchanged). Interestingly, the etiological invariance of CP onset also appeared to hold when the analysis of pain intensity was restricted to CP affected subjects; in this respect, due to the small number of pairs, we could only qualitatively explore the pattern of twin correlations by zygosity, which still showed similar genetic and environmental variance components for CP intensity (now defined without the “no pain” category) and occurrence. Therefore, our data suggest that genetic and environmental factors may affect, to comparable extents, individual differences both in the liability to CP onset and, once the condition has become clinically manifest, in the degree of pain intensity.

Our result concerning the non-additive nature of genetic influences for both CP occurrence and intensity is novel in the field of twin research on CP-related traits. At the same time, we should recognize that, based on genetic theory [26], it is considered unlikely that genetic influences on a specific trait will be entirely non-additive in origin, and that additive effects will not be involved. Thus, it seems safe to conclude that the genetic effects emerging in our study hold at least a non-negligible, non-additive, possibly epistatic component. Genetic epistatic effects involve mechanisms of interaction between gene variants at different loci. These mechanisms are not totally surprising in light of the high complexity of the studied CP phenotypes, especially under the wide definition we adopted. Recent studies have already provided some evidence of the involvement of gene–gene interactions in the molecular basis of certain CP syndromes. For example, a case-control study of temporomandibular disorders observed epistatic interactions between the catechol-O-methyltransferase (COMT) rs4680 polymorphism and polymorphisms in guanosine-5-triphosphate cyclohydrolase 1 (GCH1) and estrogen receptor 1 (ESR1) [27], while a cohort study on the contribution of hypothalamic–pituitary–adrenal (HPA) axis genes to vulnerability to post-traumatic pain found significant epistatic interactions between the corticotropin-releasing hormone binding protein (CRHBP) and FK506 binding protein 51 (FKBP5) [28]. In addition, a cross-sectional exploratory study on gene–gene interactions affecting the development of post-surgical pain and opioid consumption provided initial evidence that the interaction of the A118G single-nucleotide polymorphism (SNP) of opioid receptor mu 1 (OPRM1) with four COMT SNPs affects both postoperative pain and response to opioids [29]. Of note, the understanding of gene–gene interactions involved in responses to opioids may be of relevance as pain intensity might also be indirectly affected by genetic factors implied in both opioid-induced analgesia and side effects. Our findings may encourage further research aimed at identifying specific interactive dynamics that involve genetic factors of relevance for CP liability and associated phenotypes.

The predominant individual-specific environmental influences that were detected in our study may be of value to public health and may strengthen the need for primary and secondary prevention activities aimed avoiding CP occurrence in at-risk individuals and decreasing pain intensity in CP affected subjects. Relevant stakeholders have already provided important recommendations, proposing to heighten awareness concerning pain, its risk factors, and health consequences, especially in specific subgroups (e.g., workers, elders) and to promote healthy lifestyles, including exercise, diet and nutrition, stress management, self-care, and appropriate health-seeking behaviors [30]. Specific guidelines also exist for conditions such as lower back pain, as well as calls for physical exercise aimed at the prevention of sick leave and pain flare-ups and for education about back problems based on biopsychosocial principles [31]. Other pain conditions may, however, require different actions: for preventable pain syndromes such as post-herpetic neuralgia, specific efforts could be made for public health campaigns on vaccinations, while reasonable accommodations in the workplace or psychological interventions (e.g., coping skills, activity pacing) can be helpful for musculoskeletal pain prevention [32].

It should also be emphasized that the genetic and environmental components of CP occurrence and intensity were found not to be moderated by sex or age. Therefore, given the increasing prevalence of CP with age, this finding draws attention to possible age-related differences in gene–environment interaction, namely a higher genetic responsiveness to environmental stressors and triggers in older than in younger adults. Furthermore, the invariant etiological architecture of CP across sex and age may provide the rationale for combining cohorts with different demographic characteristics, at least among the Italian population, in order to have a higher statistical power for hypothesis testing. More importantly, if population-specific effects could be excluded at least between countries with similar cultural contexts, then demographically heterogeneous samples could also be combined across twin registers from these countries to investigate CP-related hypotheses on very large cohorts of twins. To this end, information from the Italian twin sample is currently being integrated with data from the Murcia Twin Registry, obtained with a similar survey and with the same CP questionnaire, and possible etiological heterogeneity is being tested to evaluate the feasibility of combined analyses.

A major weakness of this study is the sample size, which, though respectable, was limited by the categorical nature of the variables, and did not allow reliable testing for sex differences in genetic and environmental effects, or for age differences, by exploring these effects in different age groups. Furthermore, with this sample size, it was considered over-ambitious to apply a multivariate design to incorporate information on CP comorbidities such as depression symptomatology or sleep problems, which were also collected for the study subjects. This limitation could be overcome through the above-mentioned collaboration between the Italian and the Murcia Twin Registries. Another limitation concerns the ethnic background of the study sample, which was characterized by adult twins of Caucasian origin only. Therefore, the findings of our study cannot be directly extended to other ethnic groups.

## 5. Conclusions

This is the first twin study exploring the etiological bases of CP as a wide phenotype and its intensity in a non-clinical population of both male and female subjects using a comprehensively validated self-report questionnaire. Our results show similar etiologies for CP occurrence and intensity, with moderate genetic influences and large individual-specific environmental effects, and no evidence for age- or sex-moderation. Genetic influences are likely to encompass epistatic gene–gene interactions that should be further unraveled and quantified. The environmental effects draw attention to prevention opportunities that have currently only partially been exploited.

## Figures and Tables

**Table 1 medicina-58-01522-t001:** Socio-demographic and chronic pain characteristics of the study sample.

Characteristics	Descriptive Estimates
**N twins** (N intact pairs)	1524 (762)
**Age** (mean (SD); range)	39.2 (15.1); 18–82
**Sex** (% males)	33.7% (N = 514)
**Zygosity**	
MZM	18.0% (N = 274)
MZF	39.1% (N = 596)
DZM	7.9% (N = 120)
DZF	18.9% (N = 288)
DZMF	15.2% (N = 232)
UZ	0.9% (N = 14)
**Education** (N = 1514)	
Primary/Secondary school	5.0% (N = 76)
High school	38.8% (N = 588)
Professional school	4.2% (N = 63)
Three-year university degree	14.7% (N = 223)
Five-year university degree	27.6% (N = 418)
Post-graduate specialization	9.6% (N = 146)
**CP occurrence** (Yes/No, N = 1517)	24.1% (N = 365)
**CP intensity** (N = 361)	
Very mild	3.6% (N = 13)
Mild	18.3% (N = 66)
Moderate	60.4% (N = 218)
Severe	16.1% (N = 58)
Very severe	1.7% (N = 6)
**CP origin** (N = 332)	
Surgery	2.7% (N = 9)
Trauma without surgery	39.8% (N = 132)
Tumor diagnosed by a doctor	1.5% (N = 5)
Another illness diagnosed by a doctor	35.2% (N = 117)
Poorly defined disease	20.8% (N = 69)
**CP treatment** (Yes/No, N = 362)	74.0% (N = 268)
**Benefits of treatment** (Yes/No, N = 267)	85.8% (N = 229)

Note: Number of subjects in parentheses (N) varies according to missing data for each specific item. Acronyms: CP, chronic pain; MZM, monozygotic male twins; MZF, monozygotic female twins; DZM, dizygotic male twins; DZF, dizygotic female twins; DZMF, dizygotic twins from male-female pairs; UZ, unknown zygosity.

**Table 2 medicina-58-01522-t002:** Correlation patterns and model-fitting results for CP occurrence and intensity.

CP Occurrence				
*Correlation*				
MZ (N = 430)		0.37 (0.20, 0.52)		
DZ (N = 314)		−0.004 (−0.22, 0.22)		
MZM (N = 135)		0.28 (−0.05, 0.57)		
DZM (N = 60)		−0.21 (−0.57, 0.23)		
MZF (N = 295)		0.40 (0.21, 0.57)		
DZF (N = 139)		−0.06 (−0.39, 0.28)		
DZMF (N = 115)		0.22 (−0.16, 0.56)		
*Two-group (MZ, DZ) model-fitting analysis*				
Model	−2LL	Df	AIC	
**ADE**	**1529.274**	**1488**	**−1446.726**	
ACE	1531.033	1488	−1444.967	
Proportions of variance				
	A	D	A + D	E
	0 (0, 0.43)	0.36 (0, 0.51)	0.36 (0.19, 0.51)	0.64 (0.49, 0.81)
*Five-group (zygosity and sex) model-fitting analysis*				
Model	−2LL	Df	AIC	
**ADE**	**1528.668**	**1485**	**−1441.332**	
ACE	1530.229	1485	−1439.771	
Proportions of variance				
	A	D	A + D	E
Males	0 (0, 0.46)	0.26 (0, 0.54)	0.26 (0, 0.54)	0.74 (0.46, 1)
Females	0 (0, 0.51)	0.39 (0, 0.56)	0.39 (0.20, 0.56)	0.61 (0.44, 0.80)
**CP Intensity**				
*Correlation*				
MZ (N = 427)		0.31 (0.16, 0.44)		
DZ (N = 314)		0.08 (−0.12, 0.28)		
MZM (N = 135)		0.18 (−0.09, 0.44)		
DZM (N = 60)		−0.07 (−0.44, 0.35)		
MZF (N = 292)		0.36 (0.18, 0.52)		
DZF (N = 139)		−0.07 (−0.37, 0.25)		
DZMF (N = 115)		0.34 (0.01, 0.60)		
*Two-group (MZ, DZ) model-fitting analysis*				
Model	−2LL	Df	AIC	
**ADE**	**2166.933**	**1478**	**−789.067**	
ACE	2167.348	1478	−788.652	
Proportions of variance				
	A	D	A + D	E
	0.02 (0, 0.42)	0.29 (0, 0.44)	0.31 (0.16, 0.44)	0.69 (0.56, 0.84)
*Five-group (zygosity and sex) model-fitting analysis*				
Model	−2LL	Df	AIC	
**ADE**	**2166.666**	**1479**	**−791.334**	
ACE	2167.161	1479	−790.839	
Proportions of variance				
	A	D	A + D	E
Males	0.03 (0, 0.43)	0.18 (0, 0.46)	0.21 (0, 0.46)	0.79 (0.54, 1)
Females	0.03 (0, 0.47)	0.31 (0, 0.50)	0.34 (0.17, 0.50)	0.66 (0.50, 0.83)

Acronyms: CP, chronic pain; MZ, monozygotic pairs; DZ, dizygotic pairs; MZM, MZ male pairs; MZF, MZ female pairs; DZM, DZ male pairs; DZF, DZ female pairs; DZMF, DZ unlike-sex pairs; A, additive genetic component; C, shared environmental component; D, non-additive genetic component; E, unique environmental component; −2LL, minus twice the log-likelihood; Df, degrees of freedom; AIC (= −2LL-2Df), Akaike information criterion (the model with the lowest AIC value was preferred and highlighted in bold). Numbers in parentheses are 95% CIs.

## Data Availability

The original data are electronically stored at the Istituto Superiore di Sanità and they will be made available by the authors upon reasonable request.

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
