# Peer review of "An Italian Twin Study of Non-Cancer Chronic Pain as a Wide Phenotype and Its Intensity"

_medicina, 2022, doi:10.3390/medicina58111522_

Round 1
Reviewer 1 Report
Corrado Fagnani and colleagues performed an interesting study on chronic pain in twins. The study had a good response. However, in the end, the sample was rather small. I have two major comments.
First, I could not find any information on the ethnic background of included patients. Ethnic background is important information when we discuss genetic studies.
Second, on page 4, table 1, the authors reported that 365 patients had CP. One line lower, they described paint intensity for only 361 patients. What happened to four patients?
Reviewer 2 Report
The authors report an interesting twin study on chronic pain.
Just few points.
As this is one of the first publications using this scale, I think useful to report more extensively on the The Brief Five-Item Chronic Pain Questionnaire: items, scoring and cut offs (if they exist) (lines 104-108 and 118-123).
Although CP can be an useful clinical nosography to assess the consequences of CP on health, it may be less so when looking genetic heritability. Specifically, when looking at specific pain conditions a higher heritability has been found (van Hecke et al., 2017; Gasperi et al., 2021; Magnusson et al., 2022; Hailer et al., 2021; Markulla el al. 2009; Scaini et al., 2022; Veluchamy et al., 2018).
